# Neuromorphic Tactile Edge Orientation Classification in an Unsupervised Spiking Neural Network

**DOI:** 10.3390/s22186998

**Published:** 2022-09-15

**Authors:** Fraser L. A. Macdonald, Nathan F. Lepora, Jörg Conradt, Benjamin Ward-Cherrier

**Affiliations:** 1Department of Engineering Mathematics, University of Bristol, Bristol BS8 1TW, UK; 2Bristol Robotics Laboratory, University of the West of England, Bristol BS34 8QZ, UK; 3School of Electrical Engineering and Computer Science, KTH Royal Institute of Technology, 114 28 Stockholm, Sweden

**Keywords:** tactile robotics, neuromorphic, spiking neural network

## Abstract

Dexterous manipulation in robotic hands relies on an accurate sense of artificial touch. Here we investigate neuromorphic tactile sensation with an event-based optical tactile sensor combined with spiking neural networks for edge orientation detection. The sensor incorporates an event-based vision system (mini-eDVS) into a low-form factor artificial fingertip (the NeuroTac). The processing of tactile information is performed through a Spiking Neural Network with unsupervised Spike-Timing-Dependent Plasticity (STDP) learning, and the resultant output is classified with a 3-nearest neighbours classifier. Edge orientations were classified in 10-degree increments while tapping vertically downward and sliding horizontally across the edge. In both cases, we demonstrate that the sensor is able to reliably detect edge orientation, and could lead to accurate, bio-inspired, tactile processing in robotics and prosthetics applications.

## 1. Introduction

Research in artificial tactile sensing is crucial to enable dexterous manipulation with robotic hands. The use of bio-inspired sensors and methods can also lead to advances in our understanding of biological touch and to the development of systems that can interact more seamlessly with biological systems, an important consideration for prosthetic devices.

This research utilises the NeuroTac, a neuromorphic optical tactile sensor comprising an event-based camera recording events from a compliant skin (adapted from the original TacTip [1]). The biomimetic nature of the sensor’s skin structure and data output (in the form of spikes) renders it particularly adapted to the detection of edges. We used the NeuroTac to interact with an edge of 30° sharpness through tapping vertically and sliding horizontally at different sensor roll angles. The data were processed in a 3-layered spiking neural network with unsupervised learning, with the output from the network being fed to a 3-nearest neighbour classifier to ascertain the angle of the edge relative to the sensor. This research seeks to develop biomimetic algorithms and sensors for artificial touch, leading to advances in robotic manipulation and prosthetics.

### 1.1. Neuromorphic Tactile Sensors

Tactile sensors come in all shapes and sizes. Their underlying technologies are varied and can be based on task-dependent designs. Some examples of existing tactile sensor technologies include capacitive sensors [2], piezoelectric sensors [3], piezoresistive sensors [4], quantum tunnelling composites [5], and optical sensors [1,6].

Camera-based optical tactile sensors, as we use here, have the advantages of being high-resolution and able to exploit computer vision algorithms for data processing. Our sensor is based on the TacTip [1], a well-established tactile sensor which has been applied to a wide range of tactile tasks [7,8].

Rather than use a traditional frame-based camera, the version of the sensor used in this study is neuromorphic. Neuromorphic sensors output spike-based data, which more closely mimics biological signals, and have even been used to directly interface with the human nervous system to create prosthetic eyes [9]. One of our long-term research goals is to replicate that success in artificial vision for artificial touch. Neuromorphic systems also have the advantage of being low latency, power-efficient, and operating at high temporal resolution (events are produced at the MHz range) [10].

Although other neuromorphic tactile sensors do exist, they tend to have relatively low spatial resolution [11,12], whereas the NeuroTac (Figure 1) exploits recent developments in event-based cameras [13] to produce a high spatial and temporal resolution spike-based output.

### 1.2. Tactile Sensing and Edge Detection

In order to interact with the environment through touch, an agent requires continuous information, immediate feedback and limited noise [14]. In humans, fine motor interaction with tactile stimulus by the mechanoreceptors in our fingers occurs almost seamlessly without requiring conscious cognitive agency [15]. However, current robotic systems lack the level of feedback and control required for human-like dexterous manipulation.

To achieve this level of feedback, tactile sensing technologies have been applied to tasks such as texture detection [16,17], slip detection [18,19], and object recognition [20], all useful parts of the tactile interactions necessary for dexterous manipulation.

In humans, edges and vertices are the most salient local features of a 3D shape, and are key to the identification and in-hand manipulation of complex objects [21]. Edge detection is, therefore, an important feature for artificial tactile systems within robot hands or prosthetics [22]. The current study aims to tackle this facet of tactile sensing and focuses on the discrimination of edge orientations with a neuromorphic artificial tactile sensor, the NeuroTac.

Other studies investigating edge orientation with neuromorphic tactile sensors have used different approaches, such as spatiotemporal correlation matching of spike outputs [23] or delays in a spiking neural network for coincidence detection [16]. Another recent study created a three-layer network emulating peripheral biological circuits [24]. This study distinguishes itself from those articles through our use of a high-resolution optical tactile sensor emulating fast adapting mechanoreceptor output and our focus on a shallow feed-forward spiking neural network to process spikes. We focus on a simple network to enable its future integration on light-weight embedded systems for prosthetics and take inspiration from biology, with evidence suggesting that edge orientation is processed in first-order tactile neurons at the periphery of the nervous system [25].

### 1.3. Spiking Neural Networks

The human brain processes information as trains of spikes. Although the most biologically plausible and efficient method to encode information as spike trains is still an open question in neuroscience [26], there is a general consensus that spiking neural networks (SNNs) are currently the most biologically accurate neural network models of the brain.

SNNs have been applied to a number of problems in robotics, including robot control [27] and navigation [28], and numerous applications in robotic vision, including convolutional versions of SNNs [29,30]. SNNs efficiently process sparse data, which leads to low computational and power requirements, as demonstrated by recent benchmarking efforts [31]. Here we aim to exploit these properties of SNNs and focus on a simple feed-forward network structure for the eventual integration of neuromorphic embedded systems in prosthetic and robotic hands.

A number of strategies have been proposed to implement learning in SNNs through supervised methods, which replace traditional back-propagation, such as ReSuMe [32], SpikeProp [33], or SLAYER [34]. Here, our aim is to focus on low complexity and biological plausibility. As such, we apply a synaptic weight update rule known as spike-timing-dependent plasticity (STDP). STDP is a method of unsupervised learning in which synaptic weights are strengthened or weakened based on the temporal correlation of input and output spikes in a neuron [35].

## 2. Materials and Methods

### 2.1. Hardware—The NeuroTac Sensor

The NeuroTac sensor design is based on the TacTip sensor [1], which involves a small rubber dome encompassing pins submerged in transparent silicone gel (Techsil, RTV27905) in which pin displacements convey tactile information. In contrast with the original TacTip sensor, the NeuroTac [36]) integrates an event-based camera (Figure 1), which fundamentally changes the nature of the sensor’s output and its associated algorithms for perception. The use of Dynamic Vision Sensors in the NeuroTac leads to an event-based output analogous to biological spikes.

The version of the NeuroTac developed here features a lower form factor than its predecessor [36] (≈30 mm diameter, ≈20 mm height), enabling easier integration into robotic or prosthetic hands. This miniaturisation was enabled, in part, by the use of a lower form-factor event-based Dynamic Vision Sensor (mini-eDVS) camera [37]. The mini-eDVS records local changes in luminance at the pixel level, which outputs microsecond precision events once a threshold is reached. In the NeuroTac, events are produced when the biomimetic pins within the compliant skin are displaced due to contact.

### 2.2. Experimental Setup and Procedure

The NeuroTac sensor is attached to the end of a 4-axes robot arm (Dobot Magician, M0651) with the edge stimulus fixed to an optical breadboard beneath the robot (Figure 2). The Dobot Magician has a repeatability accuracy of 0.2 mm. Therefore, we may expect some minor noise during data collection. The Dobot robot arm is controlled through a python wrapper on the ‘common robot interface’ framework developed at Bristol Robotics Laboratory (github.com/jlloyd237/cri (accessed on 30 July 2022)). Data are gathered from the sensor through iniVation’s DV Graphical User Interface (inivation.com/dvp/dvsoftware/ (accessed on 30 July 2022)), which streams events over TCP/IP to a python script. We explored two different methods of interaction between the NeuroTac and the edge stimulus: the first is a vertical tapping motion onto the edge (referred to as Tap) and the second a horizontal movement across the stimulus (referred to as Slide).

Data collection proceeds in a similar manner for both conditions. Prior to the commencement of each trial, the tip rests 2 mm above the stimulus. In Tap trials, the NeuroTac is centred in relative XY space to the centre point of the stimulus. In Slide trials, the NeuroTac is positioned central in X space but at an 8 mm displacement in Y space. In the Tap condition, the NeuroTac descends to a point 2 mm deep in Z space relative to the apex of the stimulus before returning to its resting position above, concluding the trial. In the Slide condition, the Dobot descends to a point 2 mm deep in Z space relative to the apex of the stimulus, then moves 20 mm in the Y direction, enabling the NeuroTac to slide across the stimulus before being elevated to its initial Z coordinate and then returned to its resting position. Orientations are gathered from 0–170° in steps of 10°, with 10 trials at each orientation.

### 2.3. Spiking Neural Network Architecture

Data ouptut from the NeuroTac sensor is processed through a spiking neural network. The network consists of three layers: an input layer, a hidden layer, and an output layer (Figure 3). The input layer is a direct representation of the data output by the NeuroTac, without any need for conversion to spikes due to the asynchronous event-based output of the sensor. The hidden layer implements a spike-based version of pooling, with its neurons having circular receptive fields in the input layer of a set diameter (pooling factor = 16 in the optimised network). The output layer in the optimised network consists of 100 neurons whose firing rate is used to classify edge orientation.

#### 2.3.1. Input Layer

The first layer of the network consists of 128 × 128 neurons, totalling 16,348 neurons which correspond to each pixel output of the NeuroTac sensor. The neurons are arranged in a 128 × 128 cm grid, with 1 cm grid spacing between neurons. The unit of measurement here is employed to give the user understanding of the relative position between grid-spaced layers.

The neural network operates in the millisecond range, which leads to duplicate timestamps for some of the microsecond range events output from the sensor. Implementing a refractory period ensures that these duplicates are removed from the input layer. This layer’s output consists of spike event information transmitted through a list of varying lengths denoting times of individual neuron activation.

#### 2.3.2. Hidden Layer

The hidden layer operates as a spike-based pooling layer. Neurons in this layer are spatially distributed to cover the grid of neurons in the input layer. For instance, 64 neurons in this layer would be arranged in a grid of 8 × 8, with 16 cm grid spacing. Each neuron is then assigned a circular receptive field size equal to the downsampling between input layer neurons and pooling layer neurons (for 64 neurons in the hidden layer, the receptive field diameter would be 16 cm). Note that this entails some overlap between receptive fields of neurons in the hidden layer.

The weight of synaptic connections between the input and hidden layer is fixed at w = 0.2.

#### 2.3.3. Output Layer

The output layer contains a vector of neurons ($n_{output}$ = 100 in the optimised network), each receiving input from 16–32 random neurons in the hidden layer. We randomise the initial weights to synaptic connections between the hidden and output layer in the range of *w* = 0–0.7 and apply STDP (see Section 2.5) to these weights during learning.

The spike rates of output neurons are then fed to a 3-nearest neighbour classifier to determine edge orientation.

### 2.4. Neuron Model

The neuron model used in our SNN is an adaptive exponential integrate-and-fire (IAF), as described by Brette and Gerstner [38], built using the PyNN frontend [39] to the NEST simulator [40]. This model strikes a balance between computational efficiency and the complexity of neuron behaviour.

The model is defined by the following equations:(1)$$\begin{array}{cc} \frac{CdV}{dt}= & g_{L}\left(V-E_{L}\right)+g_{L}*\Delta_{T}*exp\left(\frac{V-V_{T}}{\Delta_{T}}\right)-\\ & g_{e}\left(t\right)\left(V-E_{e}\right)-g_{i}\left(t\right)\left(V-E_{i}\right)-w+I_{e} \end{array}$$
where *V* is the membrane potential, *C* is the membrane capacitance, $g_{L}$ is the leak conductance, $E_{L}$ s the leak reverse potential, $V_{T}$ is the spike threshold, and $\Delta_{T}$ is the slope factor. The spike-adaptation current *w* is defined by
(2)$$T_{w}*\frac{dw}{dt}=a\left(V-E_{L}\right)-W$$
where a is the subthreshold adaptation.

For more details on the neuron model, we refer the reader to the original work defining it [38]. In Table 1, we include the basic parameters of the model we used, which determines its spiking behaviour.

### 2.5. Unsupervised Learning

As mentioned in the previous section, STDP is implemented between the hidden and output layers of the network—a method of optimising learning efficiency by learning patterns of correlation between pre- and post-synaptic spike events [41]. The STDP function here is denoted as
(3)$$\Delta w_{j}=\sum_{f=1}^{N}\sum_{n=1}^{N}W\left(t_{i}^{n}-t_{j}^{f}\right)$$
in which $\Delta w_{j}$ is the weight change from the pre-synaptic neuron *j*, and $t_{j}^{f}$ and $t_{i}^{n}$ represent the spike arrival times of the pre- and post-synaptic neurons, respectively. The *W* learning function of STDP is defined as follows:(4)$$W\left(x\right)=\left\{\begin{array}{cc} A_{+}\;exp\left(\frac{-x}{\tau_{+}}\right), & \mathrm{for}\;x>0\\ A_{-}\;exp\left(\frac{x}{\tau_{-}}\right), & \mathrm{for}\;x<0\; \end{array}\right.$$
where $A_{+}$ determines the learning rate for weight strengthening when pre-synaptic spikes precede post-synaptic spikes, and $A_{-}$ determines the learning rate for weight weakening in the opposite case. $\tau_{+}$ and $\tau_{-}$ are the time constants of the positive and negative portion of the STDP curve, in our network, both set to 20 ms. We also set hard bounds on the weights within the network as $w_{max}$ (which we optimise) and $w_{min}=0$.

## 3. Results

### 3.1. Inspection of Data

The NeuroTac sensor outputs raw data in the form of events analogous to biological spikes. In this section, we inspect examples of raw data for the Tap and Slide experiments and visualise the distribution of each dataset using t-distributed stochastic neighbour embedding (t-SNE).

#### 3.1.1. Raw Data

As mentioned in Section 2.1, the NeuroTac sensor outputs event-based data. The data are in the Address-Event Representation (AER) format, which contains the pixel location and microsecond timestamp of each recorded event. Converting these timestamps to milliseconds and treating each pixel as a neuron allows us to represent the sensor output as a spike raster plot akin to neurological data (Figure 4).

From this data, we can observe that tapping contact occurs approximately between 200 and 600 ms and sliding contact between 1000 and 1400 ms. The sliding spike raster plot (Figure 4b) also presents a distinct pattern, with a decrease in overall spike activity towards the midpoint of contact. This pattern is likely due to the way the NeuroTac’s internal pins are deflected as the sensor moves across the edge stimulus.

#### 3.1.2. Data Clustering

Here we perform a rapid visualisation of the clustering of our raw data for the tapping (Figure 5a) and sliding (Figure 5b) datasets using t-SNE [42]. t-SNE enables high-dimensional datasets to be reduced into low-dimensional graphical representations of the data, presenting clusters of data points that share similarities. For both datasets, the input space for t-SNE is the number of events per pixel; further parameters were set as perplexity = 20, learning rate = 400, and max iterations = 1000.

The clustering of Tap data (Figure 5a) suggests there are four distinct clusters within the data, corresponding to orientations of (i) 20–40°, (ii) 50–90°, (iii) 160–180° and 10°, and (iv) 90–150°. The orientation groupings could be a consequence of the morphology of the NeuroTac skin and its relatively low internal pin resolution.

Contrary to this, the sliding data (Figure 5b) appears to present separable clusters by angle orientation, which we would expect to lead to a more robust classification performance overall.

### 3.2. Network Optimisation

Our initial optimisation concerns the architectural structure of the SNN. We aim to maximise classification performance whilst retaining a simple network structure. We target two parameters that determine the sizes of the hidden and output layers:Pooling factor (pf): this variable sets both the degree of reduction in network population size from the input to the hidden layer and the receptive field diameter of neurons in the hidden layer (we found from initial runs that setting these values equal led to the best performance). The range of values tested for the pF parameter are found in (Figure 6)Output neurons ($n_{output}$): This parameter simply determines the number of neurons in the output layer. We vary it over the range 50–450 neurons in steps of 50.

Since the parameter ranges are limited, we apply a brute-force approach and test classification performance for every combination of the pf and $n_{output}$ variables (Figure 6a). We optimise over a subset of our datasets comprising 9 orientations and 10 trials for each orientation to enable additional optimisation iterations.

Classification performance peaks for pf = 16 (Figure 6a), which creates a hidden layer comprising 256 neurons. The number of output neurons has less of an effect on classification performance, but we set $n_{output}$ = 100 neurons as this is the smallest number of neurons that achieves maximal performance (Figure 6a).

Learning is then optimised by varying three parameters of the STDP model:$A_{+}$, $A_{-}$: these variables set the rate at which potentiation and depression of synaptic weights occur, respectively. We vary them both independently over the range [0.001–0.1]$w_{max}$: This parameter is an upper boundary for synaptic weights. We optimise it over a range of [1–6]

We optimise the STDP parameters through Bayesian optimisation using the python hyperopt package. The final optimised parameter values of the network are displayed in Table 2.

### 3.3. Network Dynamics

Here we explore the dynamics of the SNN as it processes the data. We present spike raster plots at three stages of the network: its input, after dimensionality reduction, and the output layer. We present similar data in spike count format through heat maps of the layer activity.

#### 3.3.1. Spike Raster Plots

Input layer— In the input layer (Figure 7), the greatest activity is noticeable within the central neurons. This is due to the dome-like structure of the NeuroTac, as the central region of the sensor will incur the most displacement and, therefore, the greatest activity. In the temporal domain, the input layer shows its greatest activity between ≈200–600 ms for tapping and ≈600–1200 ms for sliding; the periods of contact with the edge stimulus. For tapping, there is a noticeable drop-off in neuron activity after the halfway point of 400 ms. This could be due to the internal dynamics of the NeuroTac sensor, with pins returning to their original positions at a slower rate than their deflections as a result of pressing down on the edge stimulus. For the sliding condition, there is a pattern of circular gaps in activity, which likely are a trace of the NeuroTac’s internal pins. Pins and the gaps between them are relatively large to the mini-eDVS’s pixel resolution. As such, at certain angles of sliding, there will be groups of pixels that are not covered by the pin displacements, which lead to gaps in activity during the sliding interaction.

Hidden layer—For the hidden layer, horizontal gaps appear in the spike raster plot. This is due to the neuron number allocation occurring sequentially across the grid of topological hidden layer neurons, with neurons connected to the outer regions of the sensor displaying low activity. The hidden layer presents significantly fewer neurons than the input layer but displays a similar overall shape of spike activity for both tapping and sliding. This is due to the input and hidden layers being topographically related, with neurons connected based on their location in space. We also notice setting the synaptic weights between input and hidden layers to a fixed value of 0.2 has significantly reduced some of the noise in the input layer (for instance, spike activity before 200 ms).

Output layer—Spiking activity in the output layer is significantly lower than the input and hidden layers and no longer displays a similar pattern of activity to the input and hidden layers. This is due to the randomised connections between hidden layer neurons and their corresponding output layer neurons. In the sliding condition, there appears to be more activity in the output layer than in the tapping condition. This is a reflection of the sliding interaction creating more spikes and activating more pixels in the NeuroTac’s output.

#### 3.3.2. Weight Updates through STDP

##### Heatmaps

Input layer—The input layer heatmap for tapping visually displays an angle orientation of 120° (Figure 8). Pin displacement occurs mostly along the stimulus edge, and the areas of high activity reflect this. The circular features within the heatmaps showing a lack of spikes represent the NeuroTac’s internal pins, whose white markers only produce events at their edges as they deflect. For the sliding condition, edge orientation is less visually evident than for the tapping condition. However, the input layer heatmap shows that a larger area of the NeuroTac skin is employed throughout the data gathering procedure. Small clusters of increased activity are also noticeable in the upper right quadrant of the pin depressions. These could indicate a preferred direction of deflection of internal pins, which is most likely correlated to edge orientation.

Hidden layer—The hidden layer for both conditions demonstrates dimensionality reduction and noise reduction while maintaining the overall spatial pattern of spike activity as the input layer. Pin contours are less obvious in this layer, but the edge orientation is still clearly displayed for the tapping condition.

Output layer—The output layer is less visually informative than the other two layers, as the connections from the hidden to the output layer are no longer topological. The 10 × 10 grid shown here is, therefore, not representative of the spatial structure within the layer and simply represents the 100 output neurons that are each connected to 16 random neurons in the hidden layer. We can identify several neurons in the output layer in each condition that demonstrates high activity and is likely to be responsive to the particular edge orientations displayed here.

In order to learn the features of the raw data, the weights between the hidden and output layer are subject to STDP learning (see Section 2.5). Here we illustrate the progression of the synaptic weights between the hidden and output layers at the 1st, 10th, and 140th iteration of training data (Figure 9). The dominance of a small number of connections that reach $w_{max}$ can be detected by 140 training cycles (Figure 9, right panel). The optimisation of the STDP parameters $A_{+}$, $A_{-}$, and $w_{max}$ leads to a balance between potentiation and depression to create specialised feature detectors in the output layer.

### 3.4. Edge Orientation Classification Performance

Once the network is optimised, we split our data using an 80% train, 20% test ratio. We test the performance by feeding the spike rates of the output neurons to a 3-nearest neighbours classifier. The input to the classifier is thus a 100 dimensional vector of output neuron spike counts.

The classification accuracy for the tapping condition reaches 62.5%, and for the sliding condition, 54.2%.

As evidenced from the confusion matrices (Figure 10), errors in orientation prediction tend to be within one to two classes of the correct orientation. Average orientation errors thus give a clearer sense of the sensor and network’s performance, with an average orientation error of $4.44^{\circ }$ for tapping and $5.69^{\circ }$ for sliding.

As a method of comparison, we also performed a 3-nearest neighbour classification on the raw input data. The Tap condition’s performance was 54% accurate, with a $6.11^{\circ }$ average orientation error, demonstrating improved performance with the SNN. The Slide condition’s performance was greater on the raw data set with 83% accuracy and an average angle error of $1.67^{\circ }$. Note that the dimensionality of the data before and after being processed by the network is not equivalent (16,384 for raw data, 100 for network processed data), but these results illustrate there is a clear difference in how the network affects the tapping and sliding tactile interactions, which we discuss in more detail in Section 4.2.

## 4. Discussion

Our aim was to develop an efficient, bio-inspired system for tactile edge orientation classification. To achieve this, we developed a neuromorphic tactile sensor and connected it to a minimally complex 3-layer feed-forward spiking neural network with STDP learning applied to its final layer and output fed to a 3-nearest neighbour classifier. Our system was able to accurately detect edge orientation while tapping on (average orientation error: $4.44^{\circ }$) and sliding across (average orientation error: $5.69^{\circ }$) an edge stimulus.

### 4.1. Hardware

The internal pins within the NeuroTac are large relative to the pixel resolution of the sensor’s output, leading to large areas of the data not producing spikes (Figure 8). Modifying the length of the pins would also affect their deflection during contact. Optimizing the size and length of these internal pins through a hardware optimisation procedure could lead to increased performance on an edge orientation task. Human fingertips have a density of FA1 mechanoreceptors of approximately 141 units/cm${}^{2}$ [43], leaving us a large margin for improvement in our sensor resolution.

### 4.2. Network

The architecture of our SNN is purposefully minimal to enable efficient processing that could be run on embedded neuromorphic systems. Future work aims to implement this system on a SpiNNaker board [44] to fully exploit the spike-based nature of the processing and low computational requirements.

Subsequently to the SNN, we employed a KNN-classifier to assess the network output task performance. This return to spike counts and classical machine learning for classification allows us to test the network’s feature transformation capabilities. However, in future work, we will extend the spike-based nature of our algorithm to include classification through Hebbian supervised learning [45] and a winner-takes-all classification approach.

While maintaining simplicity within the network, additional architectures could be explored, such as liquid state machines [46] or random cluster shapes of receptive fields in the hidden layer, as has been observed in biological systems [25]. As demonstrated by the improvement in classification our network brings for tapping data and the loss of classification accuracy for sliding data, SNNs may need to be designed for specific tactile interactions. As well as network structures, in future works, we will explore alternative time-based encoding methods (rank-order coding, first spike coding, and interspike intervals) for dynamic tactile task classification.

### 4.3. Tactile Task

While the task under investigation was relatively simple in construction, edge orientation detection is both an important area of investigation in tactile robotics [22] and is crucial in the long-term goal of in-hand tactile manipulation of complex objects [21]. The results from this study serve as a foundation for biologically plausible edge orientation detection in neuromorphic tactile systems. A planned extension of the existing framework will include a spike-based control algorithm for a robot arm that could produce neuromorphic tactile servoing, a step towards neuromorphic tactile in-hand manipulation.

## 5. Conclusions

We demonstrated a neuromorphic tactile sensor and associated spiking neural network able to discriminate edge orientation through two tactile interactions: vertical tapping and horizontal sliding. Average orientation errors over a range of 0–170° were 4.44% for tapping and 5.69% for sliding. This study lays a foundation for bio-inspired edge orientation detection on embedded neuromorphic systems for robotic hands and prosthetics.

## Figures and Tables

**Figure 1 sensors-22-06998-f001:**
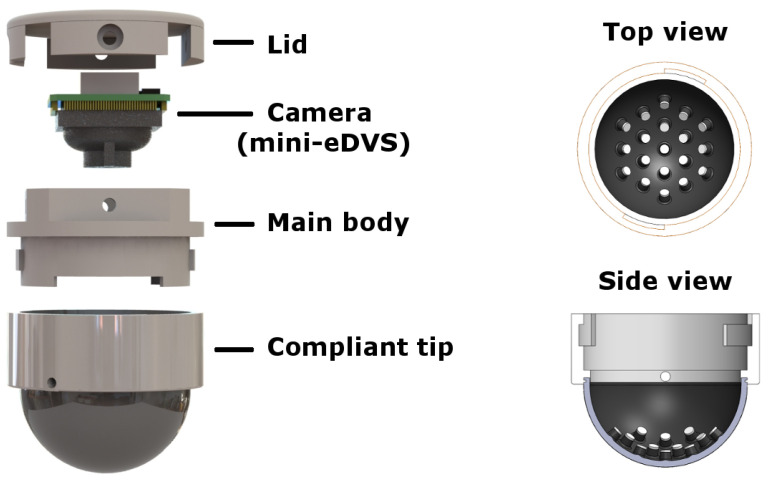
**Left Panel**: The NeuroTac—a neuromorphic version of the TacTip sensor. A lid and casing enclose the event-based camera (mini-eDVS), which, in turn, is held above a 3D-printed silicone tip with small rubber-like pins internally protruding. **Right panel**: Top and side views of the NeuroTac compliant skin, displaying the internally protruding pins that mimic the structure of human skin.

**Figure 2 sensors-22-06998-f002:**
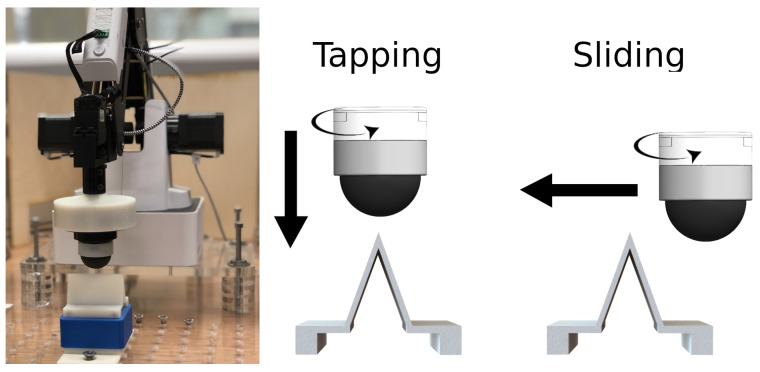
Experimental setup. **Left panel**. The Dobot Magician robotic arm with the attached NeuroTac sensor above the edge stimulus at the start of a trial. **Middle panel**. During the tapping experiment, the NeuroTac is brought vertically downward onto the edge stimulus to a depth of 2 mm, before returning to its starting position and rotating 10°. **Right panel**. During the sliding experiment, the NeuroTac is brought horizontally across the edge stimulus at a constant height of 2 mm below the apex of the edge before being returned to its start position and rotated 10° for the next trial.

**Figure 3 sensors-22-06998-f003:**
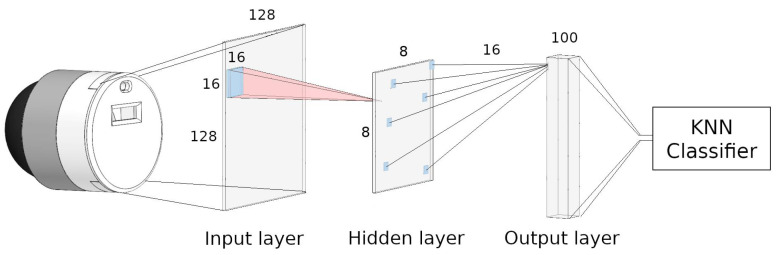
Architecture of the Spiking Neural Network. The input layer represents the captured data of the NeuroTac. Topographical pixel areas of 16 × 16 were used as inputs for the hidden layer. The output layer was then sampled again from 16 random neurons in the hidden layer. The output layer consisted of 100 neurons whose spike rate was fed into a K-nearest neighbour classifier. Note that the network structure presented here was obtained through optimisation, as described in Section 3.2.

**Figure 4 sensors-22-06998-f004:**
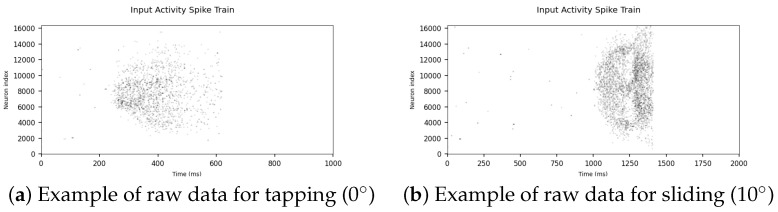
Spike raster plots of the NeuroTac output data for Tap (**a**) and Slide (**b**) at 10°.

**Figure 5 sensors-22-06998-f005:**
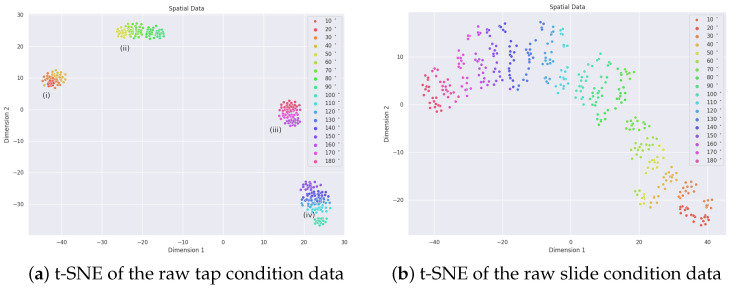
t-SNE of the spatial distribution of spikes in the raw data for tapping (**a**) and sliding (**b**) over the range of edge orientations (0–170° in steps of 10°).

**Figure 6 sensors-22-06998-f006:**
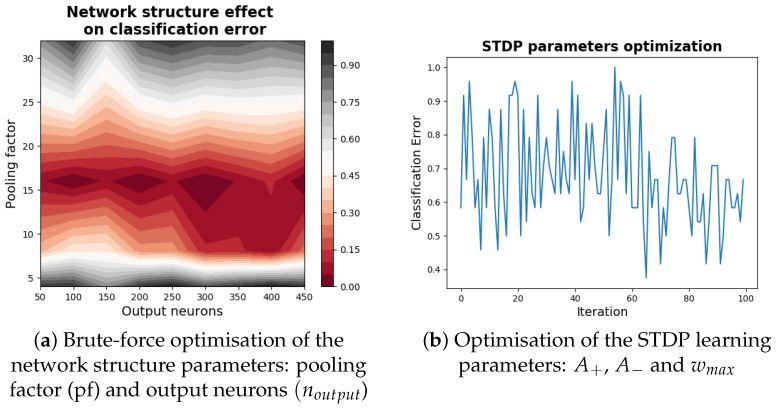
Optimisation of the SNN. (**a**) Distribution of classification performance based on pooling factor (pf) (which determines the size of the hidden layer) and the number of output neurons $\left(n_{output}\right)$. (**b**) Change in classification performance over iterations during the Bayesian optimisation of the STDP learning parameters: $A_{+}$, $A_{-}$ and $w_{max}$.

**Figure 7 sensors-22-06998-f007:**
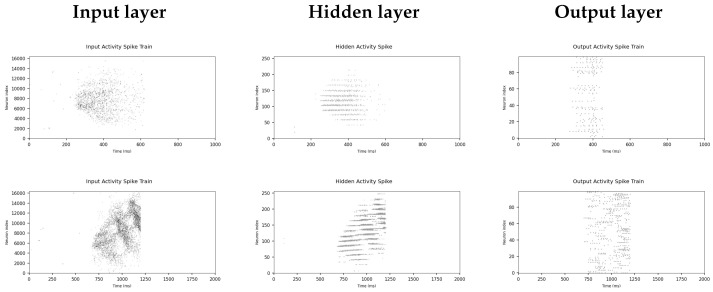
Spike raster plots representing the spiking data as they pass through each layer of the SNN. The **left panel** displays spikes in the input layer, the **middle panel** in the hidden layer, and the **right panel** in the output layer. The top row shows data for the tapping experiment and the bottom row for the sliding experiment.

**Figure 8 sensors-22-06998-f008:**
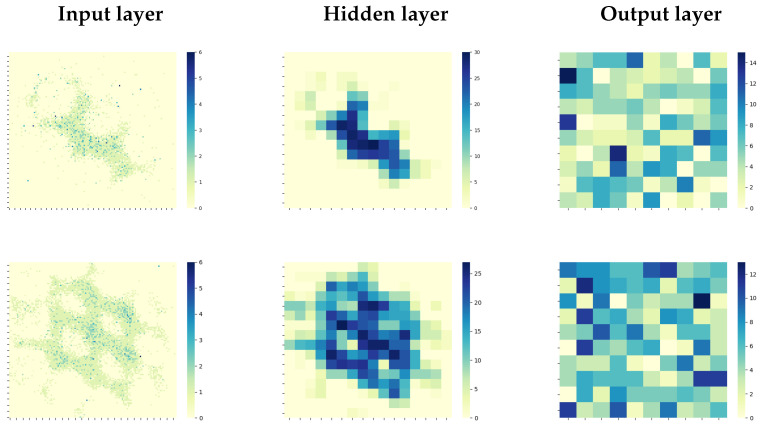
Heatmaps representing the spatial distribution of activity (degree of heat represents spike count per neuron) as data passes through each layer of the SNN. The **left panel** displays the spike activity in the input layer with a 128 × 128 layout, the **middle panel** represents the hidden layer with an optimal 16 × 16 layout, and the **right panel** the output layer with optimal $n_{output}$ = 100. The top row shows data for the tapping experiment and the bottom row for the sliding experiment.

**Figure 9 sensors-22-06998-f009:**
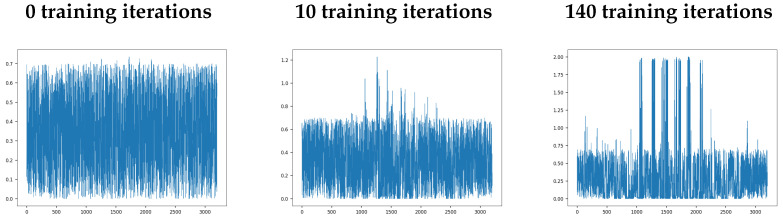
Progression of weight connections between the hidden layer and output layer as STDP operates during the training period. The *Y*-axes display the weights of synaptic connections, and the *X*-axes display synapse indexes. In this example, $w_{max}$ = 2.0.

**Figure 10 sensors-22-06998-f010:**
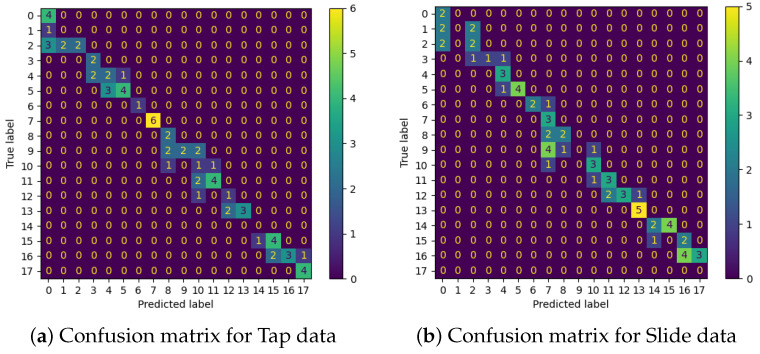
Confusion matrices for edge orientation classification performance with Tap (**a**) and Slide (**b**) over the range of edge orientations (0–170° in steps of 10°).

**Table 1 sensors-22-06998-t001:** Parameters of the IAF neuron model.

Parameter	Value
Resting membrane potential	−60 mV
Membrane capacity	1.0 nF
Membrane time constant	10 ms
Decay time of synaptic current	5 ms
Reset potential	−60 mV
Spike Threshold	−50 mv
Refractory period	5 ms

**Table 2 sensors-22-06998-t002:** Final optimised parameter values for the SNN.

Parameter	Search Space	Optimised Value
pf	4, 8, 16, 32	16
$n_{output}$	50–450 in steps of 50	100
$A_{+}$	0.001–0.1	0.036
$A_{-}$	0.001–0.1	0.073
$w_{max}$	1–6	4.75

## Data Availability

Data are available at the University of Bristol data repository, data.bris, at https://doi.org/10.5523/bris.2r2olkmnek54f2233zgufotruo.

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
