# Peer review of "Neuromorphic Tactile Edge Orientation Classification in an Unsupervised Spiking Neural Network"

_sensors, 2022, doi:10.3390/s22186998_

Round 1

Reviewer 1 Report

Abbreviation STDP in the abstract needs to be written out.

in line 85: dependant -> dependent

The experimental setup with a robot raises the question whether the position of descent was randomized? If not the reproduction of the patterns with a robotic arm motion is espected to be very accurate such that for the recognition there is no need to learn any generalization capabilities, which would weaken the results. Please specify the amount of noise in the data either introduced by purpose or due to system inherent inaccuracy.

2.3.1 & 2.3.2
What meaning has a spacial arrangement of the neurons given in cm? Is this related to a signal travel speed? Please clarify.

2.3.3
what is n_output? do you mean n_output=100 in the parentheses?

Fig. 4 is too small its hard to read on a print version.

3.1.2
What is the input space for the t-SNE? How are events represented in a high dimensional feature space?

3.2
Regarding the classification experiments you need to clarify which split of the dataset you used for training and which for test.
Text on Fig 6 is rather small too.
What is your rating of the results? Is this a good or fair result? You would need to compare it to any kind of base line. Maybe a nearest neighbour classification in the space you visualized with t-SNE could serve as a base line. If this also achieves 100% recoginition on the same test data it would be questionable why you do the enourmous effort of simulating this spiking neural network.

You should also include a learning curve that shows that the STDP optimizazion has an effect on the classification results at all. What is the performance before STDP and afterwards? I expect the netword to do a random feature transformation only while the complete recognition capabilities are in the nearest neighbour classifier.

Figure 7: Text too small again!

3.3.3
The plot in Fig.9 is not very usefull. Instead you should visualize the receptive filed of a selected output neuron before and after training as a heatmap. In this map you expect to see initially random forming to a pattern that somehow correlates to the oriented edges of the input data.

3.4
As I understand your description you put the output heatmaps so to say into the nearest neighbor classifier. This at the end  discards all effort of spiking neurons in my eyes. The benefit of a spiking neural network would be that you reach a real event detection that is independent from a sampling rate. In your setup you will not even find a detection but only a classification of previously segmented events.

You should compare the results to a direct nearest neighbor classification of a heatmap of the original sensor event data subsampled in order to support the necessity of the spiking neural network.

Reviewer 2 Report

The paper aims at experimentally investigating the feasibility of a neuromorphic system based on a tactile sensor which incorporates an event-based vision system (mini-eDVS) into an artificial fingertip (NeuroTac). Raw output data are processed through a Spiking Neural Network with unsupervised STDP learning and a 3-nearest neighbours classifier. The target application is edge orientation classification during tapping and sliding experiments performed using an ABB robotic arm. Experiments (tapping and sliding) are very simple.

The novelty of the proposed methods and achieved results is average; the authors use well established methods and tools to implement a neural classification system. Interest is in the focus on event based sensing and neuromorphic processing to target biologically plausible implementation. The applied methodology is sound and the assessment of results is convincing.

The paper presents a preliminary study on the feasibility of the system (almost a feasibility study), the results are based on very simple experiments yet they are encouraging.

The paper is well written and presentation is clear.
